# Atomistic Simulations of Plasma-Enhanced Atomic Layer Deposition

**DOI:** 10.3390/ma12162605

**Published:** 2019-08-15

**Authors:** Martin Becker, Marek Sierka

**Affiliations:** Otto Schott Institute of Materials Research, Friedrich Schiller University Jena, 07743 Jena, Germany

**Keywords:** plasma-enhanced atomic layer deposition, Monte Carlo simulation, molecular dynamics simulations, density functional theory, ReaxFF reactive force field

## Abstract

Plasma-enhanced atomic layer deposition (PEALD) is a widely used, powerful layer-by-layer coating technology. Here, we present an atomistic simulation scheme for PEALD processes, combining the Monte Carlo deposition algorithm and structure relaxation using molecular dynamics. In contrast to previous implementations, our approach employs a real, atomistic model of the precursor. This allows us to account for steric hindrance and overlap restrictions at the surface corresponding to the real precursor deposition step. In addition, our scheme takes various process parameters into account, employing predefined probabilities for precursor products at each Monte Carlo deposition step. The new simulation protocol was applied to investigate PEALD synthesis of SiO_2_ thin films using the bis-diethylaminosilane precursor. It revealed that increasing the probability for precursor binding to one surface oxygen atom favors amorphous layer growth, a large number of –OH impurities, and the formation of voids. In contrast, a higher probability for precursor binding to two surface oxygen atoms leads to dense SiO_2_ film growth and a reduction of –OH impurities. Increasing the probability for the formation of doubly bonded precursor sites is therefore the key factor for the formation of dense SiO_2_ PEALD thin films with reduced amounts of voids and –OH impurities.

## 1. Introduction

Atomic layer deposition (ALD) is a coating technology based on self-terminating reactions of gaseous precursor molecules with surface functional groups of the substrate. As shown in Figure 1, a precursor vapor (usually an organometallic compound) is introduced in the chemical reactor. It reacts with the functional groups at the surface of the substrate, until the surface is saturated (first half-cycle). Excess precursor and reaction byproducts are purged by inert gases. Next, a second precursor is pulsed in the reactor, such as an oxidizing agent like H_2_O, ozone, O_2_, or O_2_ plasma, which reacts with the remaining ligands of the precursor material (second half-cycle). The reactor is purged again. The ALD process is continued by repeating the above pulsing and purging steps, which constitute the ALD cycle, until the desired film thickness is reached. These reactions depend only on the abundancy and variety of the surface functional groups for a specific precursor, and not on the shape of the substrate. Hence, the ALD technology leads to extremely conformal coatings on highly curved [1,2,3,4,5] or nanostructured substrates. Additionally, the composition of the thin films can be controlled with atomic precision by alternating the organometallic compound for binary or more complex mixtures [1,5]. Plasma-enhanced ALD (PEALD), which utilizes oxygen plasma, is applied in order to increase the reactivity of the oxidizing agent [6]. This enables, for example, room temperature deposition of SiO_2_ thin films, which is possible only at 200 °C in typical thermal processes using O_3_ [7]. However, recently developed highly reactive precursors allow for thermal ALD processes at temperatures approaching room temperature [8,9]. The O radical species present in the plasma allow for additional tailoring of material properties [10,11]. For example, in the case of TiO_2_ coatings, PEALD films are denser and have a higher refractive index [12].

Atomistic simulations of PEALD processes are of particular interest for the fundamental understanding of the growth processes, mechanisms of material densification and crystallization, void formation, intermixing, and mechanical properties of the coatings at an atomic level [13]. However, such simulations are very challenging due to the usually significant number of competing chemical reactions and the thickness-dependent structure relaxation of the growing surface [13,14,15]. For mechanistic questions, such as reaction paths, reaction energies and barriers electronic structure approaches are the methods of choice [13]. In particular, the advent of modern density functional theory (DFT) based on accurate gradient-corrected and hybrid functionals allowed for the identification and understanding of elementary chemical reactions of a number of ALD processes [13]. For example, the understanding of SiO_2_ ALD mechanisms has proceeded through a number of computational studies employing DFT [16]. The reactivities have been investigated for a number of aminosilane precursors [17,18,19,20,21,22,23], which display low activation energies for reactions with surface hydroxyl groups. Despite the progress in the applications and development of electronic structure methods, their computational cost is still too high to reliably simulate the overall ALD growth [13,15]. This task can only be achieved by employing more approximate methods capable of extending the spatiotemporal scale accessible for simulations. The two such approaches applied for studies of ALD growth dynamics are the atomistic kinetic Monte Carlo (KMC) model and molecular dynamics (MD) [24].

KMC is a stochastic method intended to simulate the time evolution of processes that occur with priori known transition rates among states. The main advantage of the KMC method is its ability to model a broad range of time scales [24]. However, it relies on an accurate knowledge of rate constants for all elementary reaction steps. Such information is difficult to obtain by employing DFT due to the deficiencies of existing exchange–correlation functionals and requires more accurate and computationally much more demanding electronic structure methods. Another disadvantage of the KMC approach is its inability to simulate the morphology of the growing ALD film, i.e., crystalline vs. amorphous growth. Atomistic KMC simulations using DFT-derived reaction mechanisms and activation energies have been used, e.g., to investigate ALD growth of HfO_2_ [14,25].

In MD simulations, the time evolution of interacting atoms is described directly by solutions of the corresponding equations of motion [24]. Therefore, the MD method, in combination with carefully parameterized potential functions or reactive force fields, is able to describe relaxation phenomena and defects in ALD-generated films. However, the temporal scale of MD simulations (ps to ns) is very small compared to the duration of ALD pulses. This problem has been addressed by Hu et al. in MD simulations of Al_2_O_3_ ALD by separating the large time scale of surface reactions (ns to s) from the small time scale of structural relaxation (ps) [15]. Their deposition algorithm assumes that ALD reactions occur only on the active –OH groups on the growing surface and that the products of the metal precursor pulses can be fully hydroxylated, i.e., –Al(CH_3_)_2_ is fully converted to –Al(OH)_2_. The direct (large time scale) simulations of surface reactions are replaced by a deposition algorithm. This starts with a hydroxylated surface and randomly picks one of the available surface –OH groups for the deposition of the ALD product (–Al(OH)_2_). Approximate steric and overlap restrictions with neighboring atoms are checked and, if satisfied, an H atom in the selected –OH group is replaced by –Al(OH)_2_. After each deposition step, the structure is relaxed using MD simulation. Unlike KMC, this MD-based approach enables us to study the thickness-dependent evolution of the microscopic structures of ALD layers. In addition, the influences of operating parameters, such as precursor type, temperature, external fields, initial surface structure (crystalline vs. amorphous), number and distribution of –OH, etc. on the ALD process can be investigated at the atomic level.

Due to a high computational cost, MD simulations of large systems are rarely performed using electronic structure methods. Instead, more approximate approaches, such as interatomic potential functions or force fields, are used. However, the applicability of many interatomic potentials is restricted to one element oxidation state and a small number of polymorphs [26]. A relatively new and promising trend in the development of simulation methods for nanomaterials is the use of so-called reactive force fields, such as the ReaxFF approach of van Duin and co-workers [27]. This has been applied to a wide range of materials including amorphous and crystalline SiO_2_ [28,29] and Al_2_O_3_ [30], yielding very good agreement with experimental data. Although parameterization of ReaxFF is relatively complex, it is very powerful since it allows for an accurate description of the chemistry of nanostructures with a computational cost far below that of quantum mechanical methods. In our recent study, we demonstrated that the structure of amorphous SiO_2_ could be accurately described at the ReaxFF level by employing relatively small semi-amorphous periodic models [31]. A simulation cell containing only 64 SiO_2_ units yielded a structure that properly reproduced experimental structural parameters, mechanical properties, and IR spectra of bulk silica, using subsequent refinement at the DFT level.

In this article, we present an atomistic simulation scheme for a PEALD process that combines the Monte Carlo (MC) deposition algorithm and structure relaxation using MD. Our scheme is based on the one proposed by Hu et al. [15]. However, in contrast to their implementation that directly deposits hydroxylated products of the second ALD half-cycle, we use a real, atomistic model of the precursor deposited in the first half-cycle. In addition, for the MC deposition step, our approach introduces site occupation probabilities for precursor reaction products. The performance of our simulation scheme is demonstrated for PEALD synthesis of SiO_2_ thin films, using a bis-diethylaminosilane (BDEAS) precursor with the structural formula H_2_Si(NEt_2_)_2_, Et = C_2_H_5_. We show that the resulting structure of the SiO_2_ coating strongly depends on the occupation probabilities for precursor products. Densification of the SiO_2_ film and reduction of –OH impurities are observed for increasing occupation probability of doubly coordinated surface sites.

## 2. Method and Implementation

### 2.1. PEALD Simulation Procedure

Figure 2 shows the implemented protocol for simulation of the first PEALD deposition half-cycle. In contrast to the original scheme of Hu et al. [15], it used the MC method to simulate the full precursor deposition process. In the first step of each precursor deposition half-cycle, the surface –OH groups were enumerated. Then, during each MC step, a free surface –OH group was randomly selected. The surface was scanned for neighboring free –OH groups and, depending on their number and precursor type, a MC move was performed that creates one of possible precursor products. This part is also different than in the original scheme of Hu et al. [15], where direct deposition of the product of the second half-cycle (i.e., –Al(OH)_2_) was performed. Steric hindrance and overlap restrictions at the surface corresponding to the real precursor deposition step were accounted for only in an approximate way, by defining an exclusion zone around each successfully deposited –Al(OH)_2_ group. In contrast, our approach employs a real, atomistic model of the precursor. After each attachment step, the structure was relaxed, either by local structure optimization or by short simulated annealing using molecular dynamics (MD). The energy of the chemisorption was calculated and used to calculate the acceptance criterion employing the usual Metropolis–Hastings rule [32]. The procedure was repeated until all enumerated surface –OH groups were either reacted or excluded from MC moves due to steric hindrance and overlap with chemisorbed precursor molecules. Alternatively, a predefined concentration of surface –OH groups can be left unreacted. The final precursor monolayer structure was relaxed and equilibrated using MD. The second ALD half-cycle was modeled by simply replacing all organic surface groups with –OH groups, followed by relaxation and equilibration using MD. This corresponds to full conversion of the products of the precursor pulse to surface hydroxyl species. Such an approach is particularly well suited for simulating ALD growth of systems with primary reactions that have relatively low activation energies [14]. It has been shown that, unlike thermal ALD, PEALD can be efficiently performed even at room temperature [6,7], justifying the assumption of low activation energies. It has also been shown by calculations that oxygen plasma can quantitatively oxidize organic surface precursor species [17].

The selection of created chemisorbed precursor products depends on predefined occupation probabilities which remain constant during the simulated thin film growth. For example, considering a precursor yielding two possible chemisorbed precursor products **P1** or **P2** requires the corresponding occupation probabilities $\left\{p_{P1},p_{P2}\in \left[0,1\right]\right\}$, with
(1)$$p_{P1}+p_{P2}=1$$

This approach can be easily generalized to a set of N products {**P1**, **P2**, …, **P*N***} with occupation probabilities $\left\{p_{P1},p_{P2},\ldots ,p_{PN}\in \left[0,1\right]\right\}$ fulfilling the condition
(2)$$p_{P1}+p_{P2}+\ldots +p_{PN}=1$$

### 2.2. Implementation

The implementation of our PEALD simulation protocol used the Python programming language. It was designed in a modular way employing the Atomic Simulation Environment (ASE) [33], which was used for storing and manipulating structure models as well as for the reading and writing of all necessary input and output files. Structure relaxation and simulated annealing steps were accomplished employing the General Utility Lattice Program (GULP) [34]. The Python library ASE provided a built-in GULP interface which enabled single-energy calculations, structure optimizations, and MD simulations.

## 3. Test Application—PEALD of SiO_2_

As a test, the PEALD simulation protocol was applied to investigate the synthesis of SiO_2_ thin films using a bis-diethylaminosilane (BDEAS) precursor with the structural formula H_2_Si(NEt_2_)_2_, Et = C_2_H_5_. BDEAS belongs to the class of aminosilanes widely used in ALD, and it has been successfully applied for improving surface characteristics in PEALD of SiO_2_ [10,35]. It has been shown in [17] that Si precursors with amino ligands can dramatically lower the activation energies for reactions of aminosilane with surface hydroxyl groups, making BDEAS-based PEALD a very good case for testing our simulation protocol. 

### 3.1. Computational Details

#### 3.1.1. Model Systems

Hydroxylated SiO_2_ substrate and deposited precursor products are modeled using two types of cluster models of two neighboring –OH groups and a two-dimensional (2D) periodic surface. Figure 3a,b shows both cluster models. The smaller, flexible cluster model has the composition Si_2_O_7_H_6_, while the bigger and more rigid cage cluster model has the composition Si_8_O_14_H_8_. The 2D periodic substrate model shown in Figure 3c is a hydroxylated α-quartz (0001) surface containing 216 atoms, with the unit cell composition Si_48_O_120_H_48_ and lattice parameters a = 14.778 Å, b = 17.077 Å, γ = 90.081°.

#### 3.1.2. Methods

All DFT calculations employed our implementation of Kohn–Sham DFT for molecular and periodic systems [36,37,38,39,40] within the TURBOMOLE program package [41,42]. The Perdew–Burke–Ernzerhof (PBE) exchange–correlation functional [43], triple zeta valence plus polarization (def2-TZVP) [44] basis sets, and Grimme dispersion correction (DFT-D3) [45,46] were used. MD and MC simulations employed the ReaxFF reactive force field [27] using GULP-provided combined force field parameters [27,47,48].

### 3.2. Surface Reactions and Plasma Pulse Model

Figure 4 shows the possible elementary surface reactions for PEALD synthesis of SiO_2_ thin films using a BDEAS precursor. The first half-cycle starts with the deposition of BDEAS on the hydroxylated SiO_2_ surface and its reaction with –OH groups. There are two possible reactions (1 and 2a in Figure 4) that yield the organic precursor bonded to one or two surface oxygen atoms, denoted as **P1** and **P2**, respectively. Reaction 2b converts **P1** to **P2**. After the plasma pulse, **P1** and **P2** lead to a surface Si atom with three or two –OH groups, denoted as **S1** and **S2**, respectively. Alternatively, **S1** can be converted to site **S2** by reaction 3, which releases H_2_O. We have concentrated on reactions 1, 2a, 2b, and 3, since calculations indicate that oxidation of the organic precursor by oxygen radical species does not alter bonds between the precursor Si atom and the surface [17].

### 3.3. Reaction Energies

As the first step, we investigated the energetics of the elementary reactions using DFT and the cluster models of two neighboring OH groups on a hydroxylated SiO_2_ surface, shown in Figure 3a,b. The aim of the cluster studies was to obtain insights into the intrinsic energetics of the reactions without the spatial and steric constraints present at silica surfaces. All three reactions (1, 2a, and 2b) of the precursor with the cluster model in Figure 3a are exothermic, with reaction energies $\Delta E_{r}$ of −58.3, −64.3, −6.0 kJ/mol, respectively. Reaction 3, converting **S1** to **S2,** is slightly endothermic, with an $\Delta E_{r}$ of 37.5 kJ/mol. For the bigger cluster model in Figure 3b, the calculated reaction energies $\Delta E_{r}$ of reactions 1, 2a, 2b, and 3 are –67.0, –30.5, +36.5, and +63.0 kJ/mol, respectively. In the next step, we investigated the energetics of the same reactions on a hydroxylated α-quartz (0001) surface (Figure 3c). The calculated reaction energies $\Delta E_{r}$ were −63.0, −18.4, +44.6, and +60.9 kJ/mol for reactions 1, 2a, 2b, and 3, respectively. The value for exothermic reaction 1, involving only one surface OH group and yielding site **P1,** is very similar for all three models shown in Figure 3. Reaction energies for reaction 2a increase with increasing rigidity of the structure, while remaining exothermic. In contrast, reaction 2b is slightly exothermic for the small, flexible cluster model, but becomes endothermic for the structurally rigid models of the bigger cluster and surface. Reaction 3 is endothermic for all models, but reaction energies increase with increasing rigidity of the structure. Thus, reactions 1 and 2a are always energetically favorable. In contrast, reactions 2b and 3, transforming **P1** to **P2** and **S1** to **S2**, respectively, are energetically unfavorable in a rigid atomic environment. This can be attributed to constraints of the surface on the one hand, since bringing two surface oxygen atoms close enough to create sites **P2** and **S2** is connected with energy penalty. On the other hand, this demonstrates the high stability of the Si–O bonds, especially in the case of **S1**. 

### 3.4. PEALD Simulations

The cluster and periodic calculations demonstrate that the actual amount of precursor deposition products **P1** and **P2** as well as the plasma products **S1** and **S2** formed at each PEALD cycle will be influenced by the actual surface structure and the availability of neighboring –OH groups. In addition, temperature, pressure, and concentration of precursor species, and plasma energy are expected to play an important role. Accurate simulations including all these parameters would be a very complex task. As described earlier, our scheme takes the process parameters into account in a simplified way, employing predefined occupation probabilities at each deposition MC step. The probabilities for the creation of **P1** (precursor bonded to one surface O atom) and **P2** (precursor bonded to two surface O atoms) are $p_{P1}$ and $p_{P2}=1-p_{P1}$, respectively. The value $p_{P2}=1$ means that each Monte Carlo step attempts to create **P2**, and **P1** is created only if **P2** cannot be formed due to missing neighboring surface –OH groups or steric constraints. In contrast, $p_{P2}=0$ means that no **P2** is directly created during MC moves. As shown in Figure 5, these probabilities have significant impacts on the structures of PEALD layers. The simulated structures of SiO_2_ deposited for probabilities $p_{P2}=$ 0 ($p_{P1}=$ 1), $p_{P2}=$ 0.5 ($p_{P1}=$ 0.5), and $p_{P2}=$ 1 ($p_{P1}=$ 0) clearly demonstrate the densification of the growing SiO_2_ layer with increasing $p_{P2}$. For $p_{P2}=$ 0, each precursor molecule is allowed to react with only one surface –OH group, yielding **P1**. The plasma pulse converts **P1** to **S1**, with three hydroxyl groups connected to the deposited silicon atom. This increases the number of hydroxyl groups compared to the initial state, thus favoring the creation of voids, –OH impurities, and amorphous growth of deposited films. In contrast, for $p_{P2}=$ 1, after the first ALD half-cycle, the maximum possible number of **P2** precursor sites is created. The plasma pulse converts **P2** to **S2**, with two hydroxyl groups connected to the deposited silicon atom. In this case, the number of hydroxyl groups remains unchanged after the deposition step compared to the initial surface state, and, in combination with compact Si–O–Si bridges, it results in a dense film growth. Figure 6 shows the average growth per cycle (GPC) and mass densities of deposited SiO_2_ thin films obtained by 30 simulations of seven PEALD cycles for each of $p_{P2}=$ 0.0, 0.25, 0.5, 1.0. Virtually identical results were obtained for MD simulations at temperatures ranging from 300 to 1000 K at a time scale of up to 10 ps. GPC and mass densities were basically converged after seven simulated PEALD cycles. Decreasing GPC and increasing density were observed for increasing $p_{P2}$, confirming the observed densification of thin films shown in Figure 5. The very low values for GPC and density in the case of $p_{P2}=$ 0 are mainly due to the strong steric effects when only creation of singly coordinated **P1** and **S1** is allowed and creation of **P2** and **S2** is completely excluded. This confirms again the key role of doubly coordinated surface sites for the densification of SiO_2_ PEALD thin films. 

The actual reaction conditions were accounted for only in an approximate way in our occupation probability model. For instance, possible precursor desorption due to the substrate heating was not directly simulated. Possible influences from precursor temperature and concentration were not directly simulated, since our MC scheme deposits one precursor molecule after the other, disregarding the actual reaction path. However, we expect that our model can describe plasma energy effects, since the presence of plasma ions in the second half-cycle may affect reactions 2b and 3 (Figure 4), and therefore the occupation probabilities of doubly coordinated products **P2** and **S2**.

Our simulation scheme has been tested for BDEAS as a bis-aminosilane precursor containing two amino ligands. For mono-aminosilane precursors, we expect simulated structures resembling the case of $p_{P1}=$ 1 in Figure 5, but with denser films and less steric hindrance effects, since in this case there are no remaining precursor amino ligands after the first deposition half-cycle. Aminosilane precursors with three or four amino ligands can lead, in principle, to even higher surface densification, since they can increase the number of compact Si–O–Si bridges. However, the higher number of amino ligands favors steric hindrance effects, and triply and quadruply occupied surface sites strongly reduce the availability of surface –OH groups, thus inhibiting the overall film growth. In summary, the size and number of precursor amino ligands are expected to be determining factors for the steric effects and availability of –OH groups in our MC simulation scheme.

## 4. Conclusions

In this study, we presented an atomistic simulation scheme for PEALD that combines the Monte Carlo deposition algorithm and structure relaxation using molecular dynamics. In contrast to previous implementations, our approach employed a real, atomistic model of the precursor. This allowed us to account for steric hindrance and overlap restrictions at the surface corresponding to the real precursor deposition step. In addition, our scheme took various process parameters into account in a simplified way, employing predefined occupation probabilities for precursor products at each Monte Carlo deposition step. The new simulation protocol was applied to investigate the PEALD synthesis of SiO_2_ thin films using a bis-diethylaminosilane precursor. Initial analysis of the precursor products with hydroxylated SiO_2_ surfaces employing DFT calculations demonstrated a strong dependence of reaction energies on the surface structure, in particular for the precursor bonded to two surface oxygen atoms. PEALD simulations employing our new method revealed that precursor binding to one surface oxygen atom favors amorphous layer growth, a large number of –OH impurities, and the formation of voids. In contrast, precursor binding to two surface oxygen atoms leads to dense SiO_2_ film growth and reduction of –OH impurities. Increasing the probability for the formation of doubly bonded precursor sites is therefore the key factor for the formation of dense SiO_2_ PEALD thin films with reduced amounts of voids and –OH impurities. 

## Figures and Tables

**Figure 1 materials-12-02605-f001:**
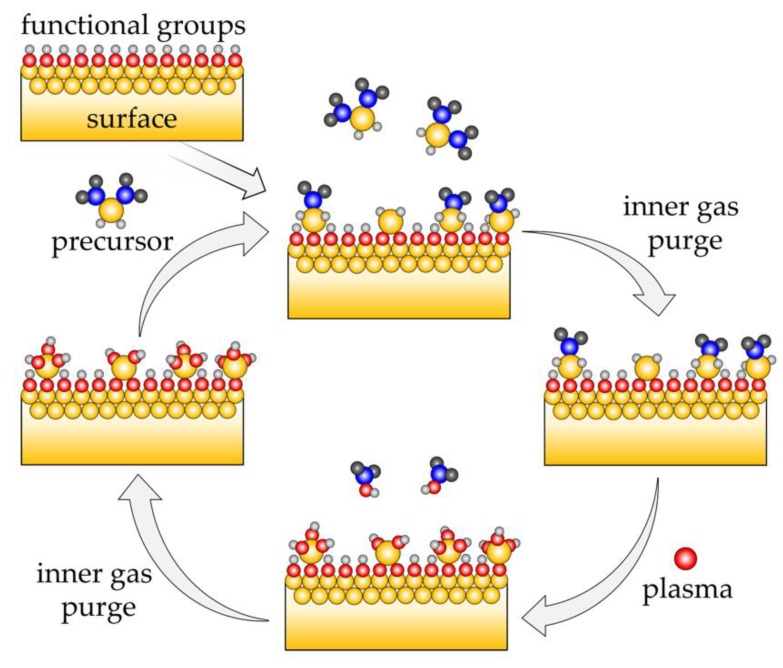
Schematic representation of a surface using plasma-enhanced atomic layer deposition (PEALD).

**Figure 2 materials-12-02605-f002:**
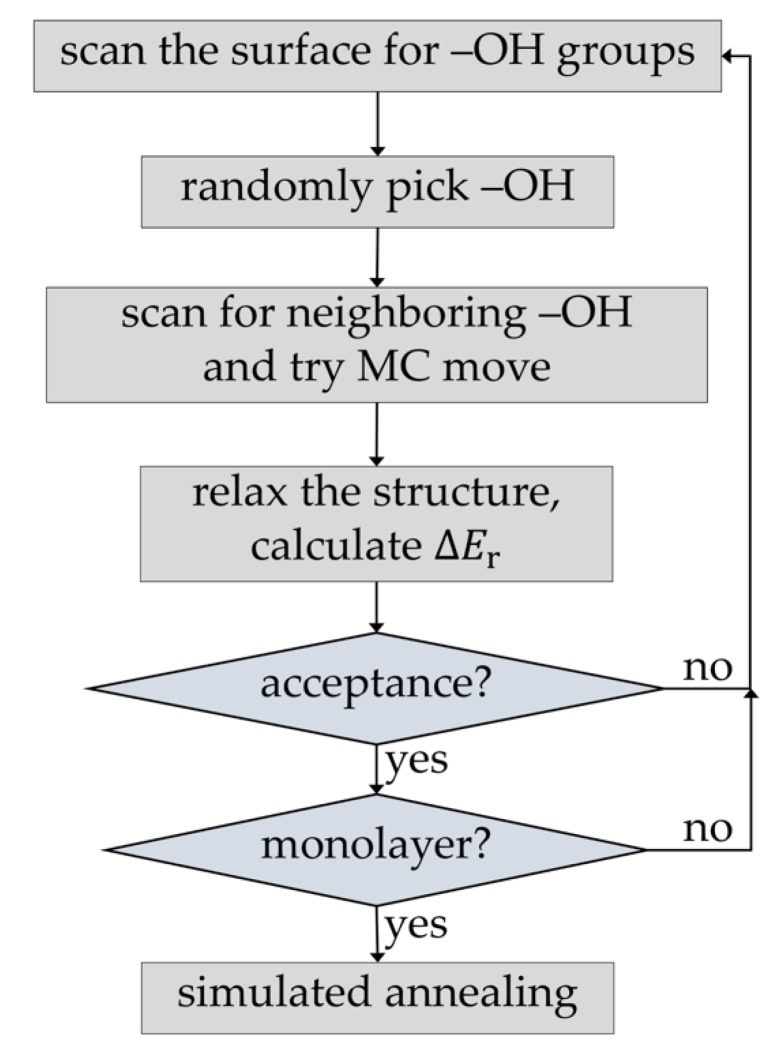
Monte Carlo (MC) simulation protocol for growth of PEALD thin films.

**Figure 3 materials-12-02605-f003:**
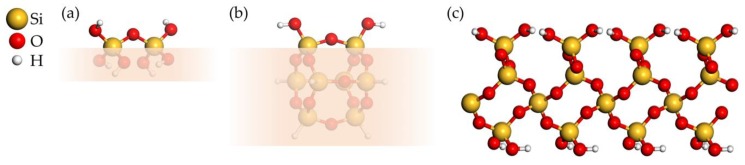
(**a**,**b**) Two cluster models of two neighboring –OH groups on a hydroxylated SiO_2_ surface. The shaded lower part indicates the “surface” part of the clusters. (**c**) 2D periodic model of a hydroxylated α-quartz (0001) surface.

**Figure 4 materials-12-02605-f004:**
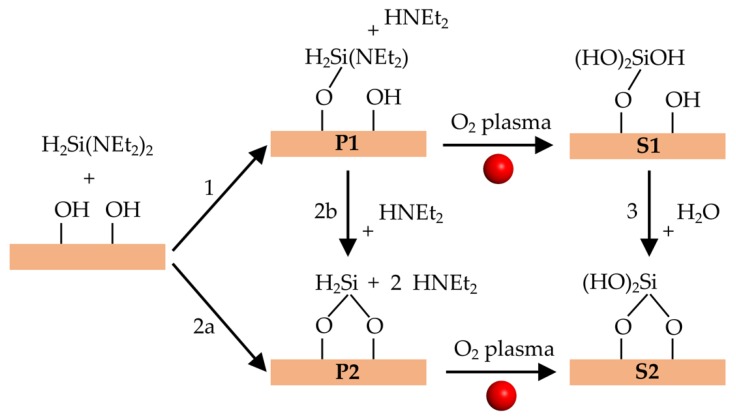
Elementary surface reactions during PEALD deposition of SiO_2_ film using a bis-diethylaminosilane (BDEAS) precursor. **P1** and **P2** are products of the precursor deposition. **S1** and **S2** are products of the plasma pulse.

**Figure 5 materials-12-02605-f005:**
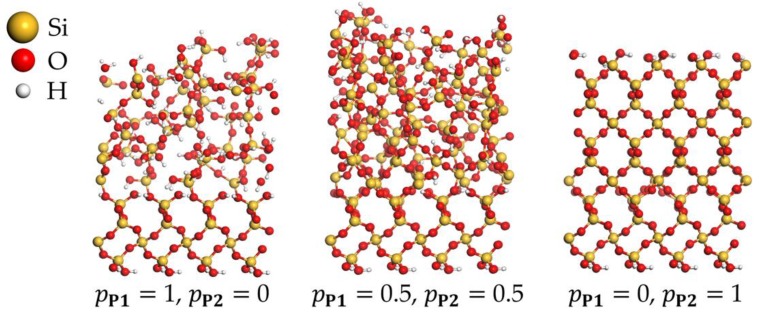
Simulated structures of PEALD-deposited SiO_2_ with different occupation probabilities $p_{P1}$ and $p_{P2}$.

**Figure 6 materials-12-02605-f006:**
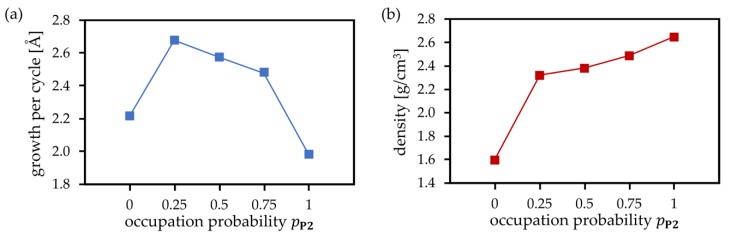
Simulated (**a**) growth per cycle and (**b**) mass density of deposited SiO_2_ films as a function of the probability $p_{P2}$.

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
