# Peer review of "Atomistic Simulations of Plasma-Enhanced Atomic Layer Deposition"

_materials, 2019, doi:10.3390/ma12162605_

Round 1
Reviewer 1 Report
The authors apply combined MC-MD approach to mimic PEALD of SiO2. Although atomistic simulations of oxide ALD is gaining high interest industrially, very few theoretical studies have been done so far, especially at “meso-scale”. Current methodology thus would potentiality have high impact toward applications. However, the scope of current study is quite limited, and several improvement are required before publication. We suggest a major revision.
1. Although authors mention “deposition of SiO2 thin films, which is possible only at 200 °C in typical thermal processes using O3”, recently developed precursors allow thermal ALD at much lower temperatures approaching RT. 1,2-bis(diisopropylamino)disilane (BDIPADS) and diisopropylaminosilane (DIPAS), for example - the discussion needs to be changed accordingly
2. Convergence in MD should be tested and provided
3. In the DFT part, the results from two different model surfaces (cluster vs. slab) are showing significant disagreement. Both models are deficient in various ways, such that cluster is too small and flexible, and quartz(0001) is too rigid and stable. It is recommended to repeat the calculation using other models that generate more consistent results by itself and also with the actual SiO2 surface under reaction conditions.
4. “Pp1 vs. pp2 ratio”, which is the key parameter here, can be strongly affected by physical/chemical conditions during deposition, such as temperature and plasma power. It is highly recommended to discuss possible connections between such parameters-“the ratio”-structure of SiO2.
5. The authors only tested a bis-aminosilane as Si precursor, which can either bond once or twice to the surface. It will be helpful if they can discuss expected differences in atomistic structures of SiO2 originating from molecular structures of the Si precursors, such as mono-aminosilane or even divalent silylene versus bis-aminosilane.
Author Response
Point 1: Although authors mention “deposition of SiO2 thin films, which is possible only at 200 °C in typical thermal processes using O3”, recently developed precursors allow thermal ALD at much lower temperatures approaching RT. 1,2-bis(diisopropylamino)disilane (BDIPADS) and diisopropylaminosilane (DIPAS), for example - the discussion needs to be
changed accordingly.
Response 1: We have modified the introduction following the reviewer suggestion and included references (refs [8,9]) to two studies of low temperature thermal ALD.
Point 2: Convergence in MD should be tested and provided.
Response 2: Information about the convergence of results in MD has been included on page 7.
Point 3: In the DFT part, the results from two different model surfaces (cluster vs. slab) are showing significant disagreement. Both models are deficient in various ways, such that cluster is too small and flexible, and quartz(0001) is too rigid and stable. It is recommended to repeat the calculation using other models that generate more consistent results by itself and also with the actual SiO2 surface under reaction conditions.
Response 3: Following the reviewer suggestion we have added results of DFT calculations for a larger cluster model. The manuscript has been modified accordingly on page 5, the cluster model has been added to Fig. 3. The discussion of reaction energies has been modified on page 6, section 3.3.
Point 4: “Pp1 vs. pp2 ratio”, which is the key parameter here, can be strongly affected by physical/chemical conditions during deposition, such as temperature and plasma power. It is highly recommended to discuss possible connections between such parameters-“the ratio”-structure of SiO2.
Response 4: A discussion of the connection between physical/chemical conditions (such as temperature and plasma energy) and occupation probabilities has been included on page 8, first paragraph. Inclusion of plasma energy effects has been also mentioned on the bottom of page 6.
Point 5: The authors only tested a bis-aminosilane as Si precursor, which can either bond once or twice to the surface. It will be helpful if they can discuss expected differences in atomistic structures of SiO2 originating from molecular structures of the Si precursors, such as mono-aminosilane or even divalent silylene versus bis-aminosilane.
Response 5: Discussion of the influence of the precursor structure on SiO2 thin films has been added on the top of page 8, second paragraph.
Reviewer 2 Report
The paper is focused on the computation modelling of plasma-enhanced atomic layer deposition on SiO2 surafces. The topic falls within the scope of the journal. The results are properly presented and discussed. I recommend its publication after the following minor revisions:
- The number of keywords should be reduced. Additionally, acronyms should be used instead of the full word (for instance, PEALD should be used instead of plasma-enhanced atomic layer deposition)
- The Bibliography should be extended by quoting recent articles on the use of atomistic simulation on the reactivy of SiO2 surfaces. Within this, recent papers on the comparision of computational modelling and experimental results regarding SiO2 reactivity should de presented and discussed in the Introduction.
Author Response
Point 1: The number of keywords should be reduced. Additionally, acronyms should be used instead of the full word (for instance, PEALD should be used instead of plasma-enhanced atomic layer deposition).
Response 1: We have reduced the number of keywords. Acronyms have been accordingly changed or added to the manuscript in the caption of Figure 1 (page 3), and in the conclusions on page 8.
Point 2: The Bibliography should be extended by quoting recent articles on the use of atomistic simulation on the reactivy of SiO2 surfaces. Within this, recent papers on the comparision of computational modelling and experimental results regarding SiO2 reactivity should be presented and discussed in the Introduction.
Response 2: We have added and discussed key citations of simulations and experiments on reactivity of SiO2 surfaces in the introduction on page 2. The citations have been added to the references on pages 9-10, references [16,18-23].
Round 2
Reviewer 1 Report
the authors successfully resolved all issues raised by this reviewer. English of the new added text may need some improvement.